# A general reaction mechanism for carbapenem hydrolysis by mononuclear and binuclear metallo-β-lactamases

María-Natalia Lisa [1,10], Antonela R. Palacios[1], Mahesh Aitha[2], Mariano M. González[1], Diego M. Moreno[3,4], Michael W. Crowder[2], Robert A. Bonomo[5,6,7], James Spencer[8], David L. Tierney[2], Leticia I. Llarrull[1,9] & Alejandro J. Vila[1,7,9]

Carbapenem-resistant Enterobacteriaceae threaten human health, since carbapenems are last resort drugs for infections by such organisms. Metallo-β-lactamases (MβLs) are the main mechanism of resistance against carbapenems. Clinically approved inhibitors of MBLs are currently unavailable as design has been limited by the incomplete knowledge of their mechanism. Here, we report a biochemical and biophysical study of carbapenem hydrolysis by the B1 enzymes NDM-1 and BcII in the bi-Zn(II) form, the mono-Zn(II) B2 Sfh-I and the mono-Zn(II) B3 GOB-18. These MβLs hydrolyse carbapenems via a similar mechanism, with accumulation of the same anionic intermediates. We characterize the Michaelis complex formed by mono-Zn(II) enzymes, and we identify all intermediate species, enabling us to propose a chemical mechanism for mono and binuclear MβLs. This common mechanism open avenues for rationally designed inhibitors of all MβLs, notwithstanding the profound differences between these enzymes' active site structure, β-lactam specificity and metal content.

[1] Instituto de Biología Molecular y Celular de Rosario (IBR, CONICET-UNR), Ocampo y Esmeralda, S2002LRK Rosario, Argentina. [2] Department of Chemistry and Biochemistry, Miami University, 651 E. High St., Oxford, OH 45056, USA. [3] Instituto de Química Rosario (IQUIR, CONICET-UNR), Suipacha 531, Rosario S2002LRK, Argentina. [4] Área Química General e Inorgánica, Facultad de Ciencias Bioquímicas y Farmacéuticas, Universidad Nacional de Rosario, S2002LRK Rosario, Argentina. [5] Louis Stokes Cleveland Department of Veterans Affairs Medical Center, Cleveland OH 44106, USA. [6] Departments of Medicine, Molecular Biology and Microbiology, Case Western Reserve University School of Medicine, Cleveland OH 44106, USA. [7] CARES, CWRU-VA Center for Antibiotic Resistance and Epidemiology, Cleveland OH 44106, USA. [8] School of Cellular and Molecular Medicine, University of Bristol Biomedical Sciences Building, University Walk, Bristol BS8 1TD, UK. [9] Area Biofísica, Facultad de Ciencias Bioquímicas y Farmacéuticas, Universidad Nacional de Rosario, S2002LRK Rosario, Argentina. [10] Present address: Laboratory of Molecular and Structural Microbiology, Institut Pasteur de Montevideo, Montevideo 11400, Uruguay. María-Natalia Lisa and Antonela R. Palacios contributed equally to this work. Correspondence and requests for materials should be addressed to L.I.L. (email: llarrull@ibr-conicet.gov.ar) or to A.J.V. (email: vila@ibr-conicet.gov.ar)

Carbapenems are "last resort" drugs for treating infections from multi-resistant Gram-negative pathogenic bacteria[1–3]. Their action is continuously challenged by the emergence and spread of new resistance mechanisms[4, 5]. As a consequence, infections caused by resistant microorganisms fail to respond standard treatments, resulting in prolonged illness and greater risk of death[2]. Carbapenem-resistant Gram-negative bacteria are rapidly emerging as a cause of opportunistic healthcare-associated infections, with high mortality rates[6]. This situation has led to a "global crisis" of antibiotics that is exacerbated by the lack of novel agents effective against these pathogens[2].

Resistance to carbapenems in Gram-negative bacteria is mostly due to the production of carbapenemases (carbapenem-hydrolysing β-lactamases). Metallo-β-lactamases (MβLs) are one of the largest and most efficient family of carbapenemases[7–10]. These enzymes employ Zn(II) as an essential cofactor to cleave the β-lactam ring and inactivate these antibacterial agents[7, 10]. Most MβLs are broad-spectrum enzymes that also hydrolyse penicillins and cephalosporins. These facts, together with the worldwide dissemination of MβL-encoding genes, raise an alarming clinical problem[8]. In particular, the gene coding for the NDM-1 MβL has rapidly spread worldwide, not only in clinical settings but also in the environment[11, 12]. Inhibitors developed for the serine-β-lactamases are not effective against MβLs, and specific inhibitors for MβLs are not yet available for clinical use[7, 9, 10].

The design of an efficient MβL inhibitor has been limited by the structural diversity of the different members of this enzyme family[7]. MβLs are classified into three different subclasses: B1, B2, and B3, which differ in their active site structures, zinc stoichiometry, loop architectures, and substrate profiles[7]. Most MβLs possess a binuclear active site in which two Zn(II) ions ($Zn_1$ and $Zn_2$) are bridged by a hydroxide (Fig. 1). This stoichiometry is found in B1 and in most B3 enzymes, in which $Zn_1$ is tetrahedrally coordinated to three histidine ligands (3 H site) and the bridging hydroxide[13–16]. However, the ligand set of $Zn_2$ differs: in B1 enzymes it is provided by residues Asp120, Cys221, and His263 (DCH site)[14], while in B3 MβLs it involves residues Asp120, His121, and His263 (DHH site)[15]; in both cases the ligand set is completed by one or two water molecules. Notably, in B2 and in some B3 enzymes, residue His116 is replaced by less common and weaker metal binding ligands, such as Asn or Gln, respectively, giving rise to active mono-Zn(II) MβLs with the metal ion located in the $Zn_2$ site[16–18]. These mononuclear enzymes display further functional and structural diversity: B2 enzymes only hydrolyse carbapenems[17] and display a DCH-like $Zn_2$ site[17], while the B3 MβL GOB-18 from *Elizabethkingia meningoseptica* is a broad-spectrum enzyme which is active as a bi-Zn(II) or as a mono-Zn(II) enzyme with the metal ion bound to the canonical $Zn_2$ (DHH) site present in binuclear B3 enzymes (Fig. 1)[16, 19].

This structural diversity has led to different mechanistic proposals[10], which mostly have involved a controversy about: (1) the essentiality of the different Zn(II) sites[20–23]; (2) the identity of the nucleophile, which has been proposed to be the $Zn_1$-bound hydroxide in bi-Zn(II) enzymes based on biochemical evidence[22, 24, 25], but has not been identified in mononuclear variants; (3) the identity of the proton donor to the β-lactam amide nitrogen; (4) the substrate binding mode[26, 27], and (5) the identification of mechanistic intermediates[28]. In this last regard, nitrocefin and other chromogenic cephalosporins have been useful as mechanistic probes to identify reaction intermediates[29, 30]. However, these intermediates were not detected in all cases analysed[31]. Furthermore, since B2 enzymes are exclusive carbapenemases, these MβLs cannot be effectively interrogated using these compounds. Indeed, carbapenems are the only substrates common to all MβLs[7].

Structures of enzyme-product complexes of B1 and B3 enzymes with hydrolysed carbapenems have been reported, providing structural insight into the mechanism[32–34], but the detection of reaction intermediates in carbapenem hydrolysis has been challenging due to the limited spectroscopic properties of these compounds. Based on a previous work in the model enzyme BcII from *Bacillus cereus*[28], we decided to interrogate a series of MβLs by a combined approach using rapid-mixing techniques coupled to a range of spectroscopies to compare the mechanistic differences across MβL subclasses with different active sites and metal stoichiometries. Notably, all MβLs hydrolyse carbapenems via a similar branched catalytic mechanism that involves accumulation of two productive anionic intermediates. The proposed structures for these intermediates allow us to suggest the proton donors in this mechanism. Mono-Zn(II) enzymes additionally reveal accumulation of the Michaelis complex, in contrast to bi-Zn(II) MβLs. We attribute this difference to the involvement of a metal-activated nucleophile in binuclear enzymes, which accelerates the first chemical step of the reaction; thereby suggesting that nucleophile activation in mono-Zn(II) enzymes does not involve the metal site. These findings suggest that design strategies for inhibitors active against the full range of MβLs should be based upon these common mechanistic features, overcoming the challenge posed by the structural diversity of these enzymes.

## Results

**Imipenem hydrolysis by a mono-Zn(II) B3 MβL**. The presence of two metal ions in binuclear MβLs complicates spectroscopic studies as the individual signatures of the two metal sites overlap. Thus, we initially studied mono-Zn(II) enzymes. Subclass B3 GOB enzymes have a Gln residue at position 116, replacing the usual His ligand, and thus impairing metal binding at the $Zn_1$ site (Fig. 1). This enzyme can be active either as a bi-Zn(II) or as a mononuclear enzyme with the metal ion located in the $Zn_2$ (DHH) site[16, 19]. We studied imipenem hydrolysis catalysed by mono-Zn(II)-GOB-18 under pre-steady-state conditions using a photodiode-array detector coupled to a stopped-flow device.

Substrate consumption during hydrolysis was monitored following absorbance at 300 nm ($Abs_{300 nm}$). Progress curves showed a lag phase followed by a triphasic decrease in $Abs_{300 nm}$ (Supplementary Fig. 1a). Time-resolved spectra corresponding to the full duration of the reaction were then acquired. Difference spectra showed accumulation of a species absorbing at 340 nm during the first fast phase (Fig. 2a, Supplementary Fig. 1 and Supplementary Table 1), suggesting the presence of a reaction intermediate absorbing at this wavelength. The progress curves could not be fit to any simple linear kinetic model (Supplementary Fig. 2a–f). Instead, the minimal kinetic scheme that accounted for the kinetic traces at both wavelengths required a branched pathway involving two productive intermediate species ($EI^1$ and $EI^2$), where $EI^2$ is the species absorbing at 340 nm in the difference spectra (Fig. 3 and Supplementary Fig. 2g–i).

Simulation of the time evolution of the different species in the reaction, based on the kinetic parameters of this branched model, reveals significant population of the Michaelis complex (ES) prior to accumulation of $EI^2$ (Supplementary Fig. 3a). The simulation further reveals accumulation of $EI^2$ in greater amounts than $EI^1$, in agreement with the lack of a direct observation of the absorption features of $EI^1$. Thus we next designed different experiments to characterize ES and $EI^2$, exploiting the different time frames of their maximal accumulation.

The Michaelis complex was characterized by rapid-freeze-quench mixing experiments coupled to X-ray absorption spectroscopy, which allows for monitoring of the coordination sphere of the metal site. We studied mono-Zn(II)-GOB-18 in the resting

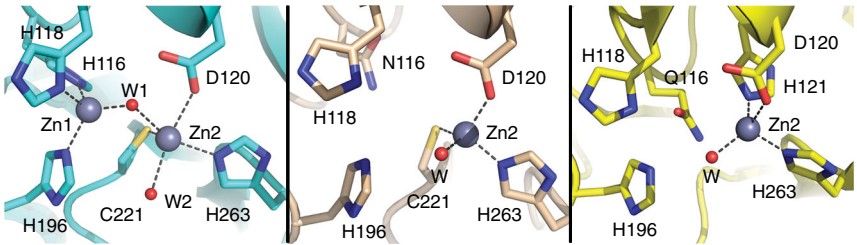

**Fig. 1** Active sites of metallo-β-lactamases. *Klebsiella pneumoniae* bi-Zn(II)-NDM-1 (B1, PDB 3spu, *left*), *Serratia fonticola* mono-Zn(II)-Sfh-I (B2, PDB 3sd9, *center*) and *E. meningoseptica* mono-Zn(II)-GOB-18 (B3, model based on PDB 5k0w, *right*). Zinc atoms are shown as *grey spheres*, and water molecules (W) are shown as *small red spheres*. Coordination bonds are indicated with *dashed lines*

state, a mixture of this enzyme and imipenem after 10 ms of reaction (where ES is predicted to be the major species), and the final enzyme-product (EP) complex (Supplementary Fig. 3b). The Zn $K$-edge spectrum for mono-Zn(II)-GOB-18 in the resting state[19] showed a first coordination shell of 4 N/O at a mean distance of 2.01 Å (including 2 His ligands; Table 1), consistent with the coordination sphere of the $Zn_2$ centre observed by X-ray crystallography (Fig. 1)[16]. After 10 ms of reaction with imipenem, the average Zn–N/O bond length increased to 2.07 Å, with the increase in distance being consistent with a larger coordination number in the enzyme-substrate complex (Table 1, Supplementary Fig. 4a and Supplementary Table 2). The first shell bond length in the product complex remained substantially longer than in the resting enzyme at 2.08 Å (Table 1, Supplementary Fig. 4b and Supplementary Table 2), suggesting that the Zn(II) ion progressed from four-coordinate in the resting state to five-coordinate in the Michaelis (ES) and product (EP) complexes. The XANES spectra (Supplementary Fig. 5) showed a dramatic increase in the white line intensity that was retained in the product complex, which was consistent with an increase in the coordination number in both ES and EP. Based on fits to the EXAFS data, we suggest this represents incorporation of the C-3 carboxylate as a ligand of $Zn_2$, with retention of the metal-bound water molecule present in the resting state enzyme.

We performed the hydrolysis reaction under conditions in which the accumulation of $EI^2$ is maximized. The difference spectrum recorded by the photodiode array at 600 ms displayed the absorption band of $EI^2$ (Supplementary Fig. 3c). These features confirm the presence of a reaction intermediate but do not provide structural information. To better characterize $EI^2$, we then studied mono-Co(II)-GOB-18. This species is active against imipenem, with similar $K_M$ but ten-fold smaller $k_{cat}$ values compared to mono-Zn(II)-GOB-18[19, 35]. The lower catalytic efficiency of the Co(II) derivative (Supplementary Table 3) and the rich spectroscopic features of the Co(II) ion, provide an opportunity to monitor changes in the metal coordination sphere during turnover and to determine the individual rate constants for formation/disappearance of intermediate species. Resting state mono-Co(II)-GOB-18 is pentacoordinate, according to the ligand field bands in the visible range[35]. After rapid mixing of imipenem and mono-Co(II)-GOB-18, time-resolved spectra showed these ligand field bands to disappear and give rise to new features, indicating that changes occurred in the coordination geometry of the Co(II) site (Fig. 2b and Supplementary Fig. 6). Later, an intense absorption band developed at 340 nm whose subsequent decay was coincident with a recovery of the ligand field features of resting mono-Co(II)-GOB-18, as evidenced by well-defined isosbestic points (Fig. 2b, Supplementary Fig. 6e and Supplementary Table 1). Kinetic traces monitoring reaction at 300 nm (reporting on substrate-bound (ES) and intermediate species), 340 nm (intermediate species), and 635 nm (ligand field bands of the Co(II) ion in the resting enzyme) at different enzyme:

substrate ratios fitted to a mechanism with two productive reaction intermediates, similar to mono-Zn(II) GOB-18 (Supplementary Fig. 7). In this mechanism, the absorption peak at 340 nm can also be attributed to $EI^2$. The smaller molar extinction coefficients at 635 nm estimated for the ES and EI complexes, compared to the values for the resting state enzyme (E), suggest an increase in the coordination number upon substrate binding and during turnover[36], in agreement with the EXAFS data for the Zn(II)-enzyme (Table 1). Recovery of the ligand field bands to intensities observed for the resting enzyme took place after consumption of this intermediate. We conclude that the metal ion in Co(II)-GOB is hexacoordinated in ES, $EI^1$, and $EI^2$. Thus, catalysis by mono-Zn(II) and Co(II)-GOB follow the same reaction mechanism, with expansion of the coordination number during turnover, differing only by the presence of an additional water molecule in the Co(II) variant.

**Imipenem hydrolysis by a mono-Zn(II) B2 MβL.** We next applied similar approaches to study imipenem hydrolysis by the mono-Zn(II) B2 lactamase Sfh-I. Transient peaks in absorbance spectra of the complete reaction time course obtained by stopped-flow absorption spectroscopy revealed accumulation of two species. First, a species with a maximum at 390 nm decayed during the early phases of the reaction and next, the absorbance at 340 nm increased and then decreased (Fig. 2c, Supplementary Fig. 8 and Supplementary Table 1). The latter resembles the spectral features observed for $EI^2$ during the hydrolysis of imipenem by mono-Zn(II)-GOB-18 (Fig. 2a), allowing us to conclude that the species absorbing at 390 nm may correspond to intermediate $EI^1$ that was derived from the fitting but could not be observed in the case of mono-Zn(II)-GOB-18.

In order to better characterize these two intermediates, we carried out a series of experiments at lower concentrations of mono-Zn(II)-Sfh-I and imipenem, with the aim of reducing the hydrolysis rate. Although the low signal-to-noise ratio at 390 nm precluded monitoring of $EI^1$ during the reaction, traces were recorded at 300 and 340 nm and analysed by simultaneous global fitting. We could not fit these data to any linear mechanism involving both intermediates. Instead, these data could be accounted for by assuming the same branched mechanism employed for mono-Zn(II)-GOB-18 (Fig. 3 and Supplementary Fig. 9). Thus, despite having different metal sites, mono-Zn(II)-GOB-18 and mono-Zn(II)-Sfh-I follow the same kinetic mechanism for imipenem hydrolysis. This, together with the similar absorbance properties of the $EI^2$ species identified for each reaction, suggests that imipenem hydrolysis by the two enzymes occurs by the same mechanism and involves the same two reaction intermediates. Simulation of the different species during hydrolysis (Supplementary Fig. 10) confirms a significant accumulation of $EI^1$, allowing us to assign the spectral features at 390 nm to this species.

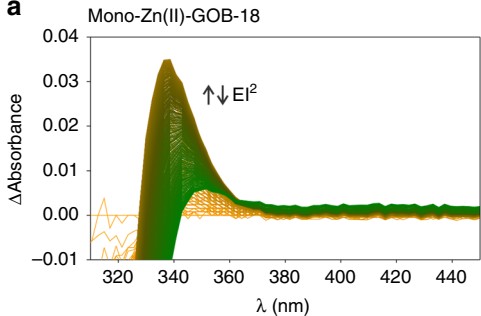

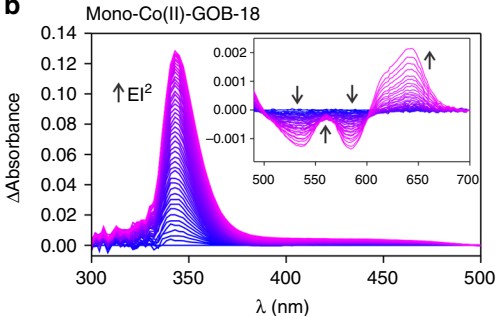

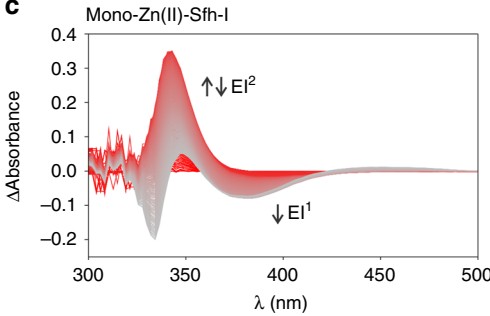

$$E + S \xrightarrow{k_1} ES \xrightarrow{k_2} EI^1 \xrightarrow{k_4} EI^2 \xrightarrow{k_5} EP \xrightarrow{k_6} E + P$$

$$EI^1 \xrightarrow{k_3} E + P$$

**Fig. 3** General reaction mechanism for carbapenem hydrolysis by MβLs. The ES complex does not accumulate in all cases and is hence depicted in a lighter colour (*grey*)

**Fig. 2** Electronic absorption spectra of imipenem hydrolysis catalysed by mononuclear MβLs. **a** Sequence of difference spectra upon the reaction of 91 μM imipenem and 51.4 μM Zn(II)-GOB-18. The reaction progresses from *yellow* to *green* spectra. The time interval covers up to 50 s. **b** Sequence of difference spectra upon the reaction of 3.5 mM imipenem and 430 μM mono-Co(II)-GOB-18. The reaction progresses from *blue* to *pink* spectra. The time interval covers from 14.7 to 26.6 s. The *inset* shows a magnification of the 450–700 nm region. **c** Sequence of difference spectra upon the reaction of 3 mM imipenem and 350 μM mono-Zn(II)-Sfh-I. The reaction progresses from *red* to *grey* spectra. The time interval covers up to 50 s

We then monitored the reaction of mono-Zn(II)-Sfh-I with imipenem using X-ray absorption spectroscopy. The XANES spectra for mono-Zn(II)-Sfh-I in the resting state, 10 ms after mixture with imipenem (when ES is the predominant enzyme species) and in the enzyme-product complex were nearly superimposable (Supplementary Fig. 5), indicating a lack of appreciable rearrangement at the metal site over the course of the reaction. This result is supported by a comparison of the EXAFS Fourier transforms (Table 1, Supplementary Fig. 11 and Supplementary Table 4). The only appreciable differences are a slight enhancement of the Zn–S scattering in the first shell of the 10-ms sample, and some complex outer shell contributions at 10-ms that disappear in the product complex, indicative of interaction with the substrate. This observation indicates that different metal site geometries can stabilise similar reaction

intermediates (such as those evidenced by absorption spectroscopy).

**Carbapenem hydrolysis by bi-Zn(II) B1 MβLs.** Hydrolysis of imipenem by bi-Zn(II)-NDM-1 under single turnover conditions revealed an increase of absorbance at 390 nm during the first 2 ms of the reaction, that then decays as another species accumulates with an absorption at 343 nm (Fig. 4a, Supplementary Fig. 12 and Supplementary Table 1). These spectral features can be attributed to the two reaction intermediates, $EI^1$ and $EI^2$, respectively, based on the similarities of their spectral features with those observed for mono-Zn(II)-Sfh-I (Fig. 2c). Since the band at higher energy (343 nm) partially overlaps with the absorption of imipenem, we fitted the time evolution of the absorption traces at 390 nm and 300 nm at different enzyme:substrate ratios. These data could be fitted to a mechanism with two productive intermediates (Fig. 3 and Supplementary Fig. 13), similar to that proposed for the two mono-Zn(II) enzymes described above. In contrast to the case for the mono-Zn(II) enzymes, the best fit was obtained when the Michaelis complex ES was omitted from the model (Supplementary Figs. 13, 14). Our data suggested that in this case formation and decay of ES occurred in the dead time of the equipment (2 ms) under all tested conditions. We also assayed hydrolysis of another carbapenem, meropenem, by bi-Zn(II)-NDM-1, and observed accumulation of two similar reaction intermediates with absorption bands at 375 nm ($EI^1$) and 336 nm ($EI^2$), respectively (Fig. 4b, Supplementary Fig. 15 and Supplementary Table 1). Overall, these results show that binuclear B1 MβLs hydrolyse carbapenems by the same branched mechanism as that described for the mono-Zn(II) enzymes.

To obtain further information on the nature of intermediates populated during carbapenem hydrolysis by binuclear MβLs, these experiments were extended to bi-Co(II)-NDM-1, which has a similar catalytic performance (Supplementary Table 3). Analysis of the hydrolysis of imipenem by bi-Co(II)-NDM-1[37] revealed formation of an intermediate species with a strong absorption band at 412 nm, matching the spectroscopic features previously reported for the hydrolysis of imipenem by bi-Co(II)-BcII (Fig. 4c, Supplementary Fig. 16 and Supplementary Table 1)[28]. The characteristic ligand field bands of the Co(II) ion in the resting enzyme (450–700 nm) disappeared upon the reaction with imipenem, giving rise to new spectral features and revealing that changes occurred in the geometry of the Co(II) sites (Fig. 4c and Supplementary Fig. 16). The time courses of the absorption at 412 (the intermediate maximum), 567 and 642 nm (d–d bands) at three different enzyme:substrate ratios were measured and fitted to different kinetic models. The progression of the intensities at 567 nm and 412 nm showed an initial increase in intensity followed by a decrease, and these features were assigned to $EI^1$. The best fits to the data were obtained using the branched mechanism involving two productive intermediates proposed for bi-Zn(II)-NDM-1 (Supplementary Fig. 17). The extinction coefficients attributed to the ligand field bands for $EI^1$ and $EI^2$ reveal that the metal sites in these intermediates have similar coordination geometries (Supplementary Fig. 17), both with higher coordination numbers than in the resting state enzyme.

**Table 1 Representative EXAFS fits for mono-Zn(II)-GOB-18, mono-Zn(II)-Sfh-I and bi-Zn(II)-BcII**

| Reaction mixture | Predominant species* | Model | Zn–O | Zn–N | Zn–S | Zn–C$_{CO_2-}$ | Zn–His | Zn–Zn |
|---|---|---|---|---|---|---|---|---|
| Mono-Zn(II)-GOB-18 resting; Fit Zn-2[19] | E | 4N (2 His) | 2.01 (7.4) | 2.01 (7.4) | | | 3.16 (20)<br>3.36 (4.6)<br>3.73 (16)<br>4.44 (22) | |
| Mono-Zn(II)-GOB-18 10 ms; Fit S.4a-1 | ES | 5 N/O | 2.07 (6.4) | 2.07 (6.4) | | | | |
| Mono-Zn(II)-GOB-18 hydrolysed imipenem; Fit S.4b-1 | EP | 5 N/O | 2.08 (9.2) | 2.08 (9.2) | | | | |
| Mono-Zn(II)-Sfh-I resting; Fit S.11a-3 | E | 3 N/O (1 His) + 1S | 2.02 (4.8) | 2.02 (4.8) | 2.29 (3.5) | | 2.82 (11)<br>3.19 (0.1)<br>4.11 (17)<br>4.47 (12) | |
| Mono-Zn(II)-Sfh-I 10 ms; Fit S.11b-3 | ES | 3 N/O (1 His) + 1S | 2.02 (2.3) | 2.02 (2.3) | 2.29 (3.0) | | 2.84 (11)<br>3.61 (0.1)<br>4.26 (2.1)<br>4.37 (1.1) | |
| Mono-Zn(II)-Sfh-I hydrolysed imipenem; Fit S.11c-3 | EP | 3 N/O (1 His) + 1S | 2.02 (3.1) | 2.02 (3.1) | 2.29 (4.3) | | 2.82 (6.8)<br>3.17 (0.1)<br>4.08 (14)<br>4.47 (14) | |
| Bi-Zn(II)-BcII resting[38] | E | 4 N/O (2 His) + 0.5S + Zn–Zn | 2.03 (6.3) | 2.03 (6.3) | 2.27 (2.6) | | 2.90 (3.1)<br>3.18 (5.8)<br>4.08 (11)<br>4.43 (15) | 3.42 (8.3) |
| Bi-Zn(II)-BcII 10 ms; Fit S.18a-6 | EI | 2N (2 His) + 2.5O + 0.5S + Zn–Zn | 1.97 (3.2)<br>2.13 (6.2) | 1.97 (3.2)<br>2.13 (6.2) | 2.3 (3.3) | | 2.91 (4.9)<br>3.13 (1.2)<br>4.20 (13)<br>4.43 (16) | 3.82 (5.3) |
| Bi-Zn(II)-BcII hydrolysed imipenem; Fit S.18b-5 | EP | 4 N/O (2 His) + 0.5S + 0.5C + Zn–Zn | 2.02 (6.0) | 2.30 (6.0) | 2.30 (7.3) | | 2.94 (3.3)<br>3.13 (0.9)<br>4.17 (14)<br>4.44 (21) | 3.51 (8.1) |

*The indicated predominant species is based on simulations using the models and parameters from the kinetic studies

We then used X-ray absorption spectroscopy to analyse hydrolysis of imipenem by the B1 enzyme bi-Zn(II)-BcII. While XAS of binuclear enzymes cannot resolve the features of individual metal sites, it provides average metal site structure, and valuable information about the Zn–Zn distance during turnover. The EXAFS of resting bi-Zn(II)-BcII (previously reported) could be fitted with an average first shell of 4 N/O and 0.5 S, including 2 His ligands per Zn(II), and a Zn–Zn separation of 3.42 Å[38]. After 10 ms of reaction with imipenem, the first shell peak in the FT diminished and broadened, being best fit with 4.5 N/O + 0.5 S per Zn(II) ion (Table 1, Supplementary Fig. 18a and Supplementary Table 5), indicating a small increase in average coordination number, and a Zn–Zn distance of 3.82 Å. The corresponding EP complex of bi-Zn(II) BcII and imipenem was best fitted with similar parameters, but a substantially shorter Zn–Zn distance of 3.51 Å (Table 1, Supplementary Fig. 18b and Supplementary Table 5). Thus, changes in the coordination geometry of the metal site are accompanied by a significant lengthening of the Zn–Zn distance in the formed intermediates.

**Identity of the reaction intermediates.** The last step in the enzyme-catalysed hydrolysis of carbapenems is expected to be the protonation of the N atom[9], giving rise to the Δ2 tautomer, that later equilibrates in solution to generate the more stable Δ1 tautomer with a 1:1 ratio of the α and β diastereomers (Fig. 5)[39]. To investigate sites of protonation in MβL-catalysed carbapenem hydrolysis, we determined the α:β diastereomer ratio for the imipenem hydrolysis products of the GOB-18, Sfh-I and NDM-1 enzymes by [1]H-NMR spectroscopy[39]. These experiments yielded values ranging between 1:5 and 1:7, i.e., with the β diastereomer in substantial excess, in contrast to the acid-induced hydrolysis (Fig. 5c). This ratio can be accounted for only by assuming that, in addition to protonation at the N atom, a diastereoselective protonation at C-2 takes place within the enzyme active site,

whose product is the Δ1β diastereomer. Since our kinetic schemes include two productive intermediates (EI[1] and EI[2]), we envisioned that these intermediates could present chemical differences that would favour either protonation at N or at C-2 (Fig. 5b).

In an effort to generate structural models for possible enzyme-bound intermediates, we inspected the available crystal structures for enzyme-product complexes of bi-Zn(II) NDM-1 with carbapenems (pdb coordinates 4eyl[32] and 4rbs). Both structures lack a metal-bound hydroxide, which would be expected to be present following N protonation by a metal-bound water molecule (Supplementary Fig. 19). The absence of a metal-bound hydroxide in the EP complexes could be instead due to C-2 protonation within the enzyme, which would then be elicited by water molecules present at other locations within the active site. We used QM-MM calculations to analyse the stability of anionic species that could give rise to these EP complexes. Simulations starting from anionic species generated from 4eyl[32] and 4rbs converged to similar Zn(II) coordination geometries (Fig. 6a, EI), with shorter N–Zn$_2$ distances than those observed in the EP crystal structures (Supplementary Table 6), consistent with the anionic nature of the intermediates. The C-7 carboxylate resulting from β-lactam cleavage bridges the two Zn(II) ions (Fig. 6a). The C2–C3 and C3–N distances of the minimized structures correspond to bond lengths intermediate between single and double bonds in both cases, suggesting a delocalized anionic structure for EI. This observation is in agreement with a previous RFQ-Resonance Raman characterization of a reaction intermediate in imipenem hydrolysis by bi-Co(II)-BcII that revealed a vibrational feature with a frequency intermediate between those of single and double bonds[28].

We also explored anionic intermediates in alternative structures containing a Zn(II)-bound water molecule, as would be expected for a mechanism involving N protonation. The minimized structures (EI$^{WAT}$, Fig. 6b) also featured a short

N–$Zn_2$ bond, with the extra water molecule bridging the two Zn (II) ions. Instead, the C-7 carboxylate moiety was found to bind $Zn_1$ only (Fig. 6b and Supplementary Fig. 20). Minimized geometries for $EI^{WAT}$ also feature delocalized single and double bonds between C2–C3 and C3–N.

Both EI and $EI^{WAT}$ feature a large charge delocalization onto the unsaturated five-membered ring and the adjacent sulphur atom (Supplementary Table 7). As a consequence, both the N and the C-2 atoms bear small partial negative charges. Thus, specific N or C-2 protonation may not be driven by the charges residing on these atoms but, instead, by the availability of a proton donor. In $EI^{WAT}$, N protonation by the bridging water is favoured (Fig. 6b), giving rise to the $\Delta 2$ tautomer. We thus explored the distribution of water molecules in the active site in EI looking for possible proton donors to C-2. Molecular dynamics simulations on EI revealed the presence of several water molecules between the hydrophilic loop L10 and the β face of the antibiotic (Supplementary Fig. 21), confirming the feasibility of diaster-oselective C-2 protonation within the enzyme active site in this intermediate. Hence, we propose that the α:β diastereomer ratio in the product of carbapenem hydrolysis by MβLs is a consequence of the existence of two alternative protonation routes (N or C-2 protonation), determined by two reaction intermediates that differ in the identity and location of the proton donor. Thus, we assign structures EI and $EI^{WAT}$ to the observed reaction intermediates.

Minimization attempts on the EP complexes generated from these intermediates gave strikingly different results. The EP lacking the Zn(II)-bound water was stable upon geometry minimization, resembling the geometries of the reported crystal structures[32, 33] (Supplementary Fig. 22 and Supplementary Table 8). In contrast, the EP with the Zn(II)-bound water was unstable in our calculations, resulting in a substantial lengthening of the $Zn_2$–N and $Zn_1$–$COO^-$ distances, which may ultimately lead to product detachment from the active site. This is consistent with our kinetic data, which show that one of the productive intermediates, $EI^1$, proceeds to product with no detectable accumulation of EP, while $EI^2$ gives rise to stable EP complexes in all studied enzymes. Hence, we conclude that $EI^1$ is $EI^{WAT}$, undergoing N protonation and giving rise to the $\Delta 2$ tautomer; while $EI^2$ is EI, which can be protonated at C-2 rendering the $\Delta 1$ β tautomer in the form of a stable EP complex (Fig. 6). The structures of the B3 lactamase SMB-1 with hydrolysed carbapenems show a $\Delta 1$ C-protonated tautomer from the β face, supporting this proposal[34]. Protonation of $EI^1$ is faster than protonation of $EI^2$ in all MβLs (Supplementary Table 9). The fact that N protonation by a metal-activated water molecule is expected to be faster, further supports the assignment of $EI^1$ as $EI^{WAT}$. Therefore, the two intermediates proceed to products through proton donors of distinct acidities which are located in different positions, and conversion of $EI^1$ into $EI^2$ involves dissociation of a water molecule from $Zn_2$.

## Discussion

Based on these results, we propose detailed mechanistic models for the hydrolysis of carbapenems by binuclear and mononuclear MβLs (Fig. 7) consistent with a unified mechanistic scheme (Fig. 3). Despite their diverse active site structures, metal content and substrate spectrum, MβLs share a branched mechanism defined by the presence of two productive reaction intermediates ($EI^1$ and $EI^2$) with similar spectroscopic features. We also provide direct evidence of changes in the coordination geometry of the metal sites.

In mono-Zn(II)-GOB, the presence of a single metal ion demonstrates the involvement of $Zn_2$ in catalysis as changes in its

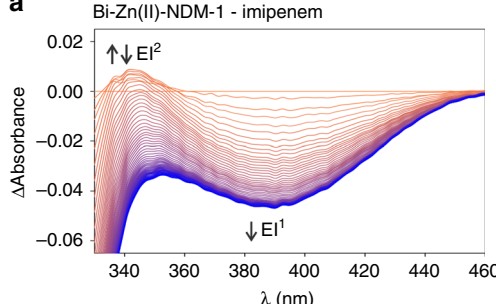

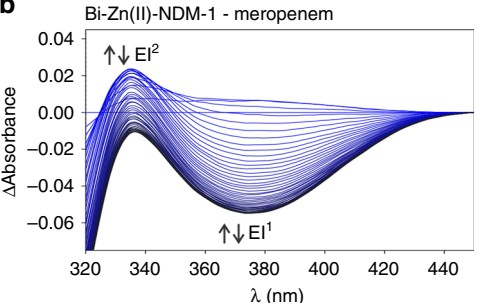

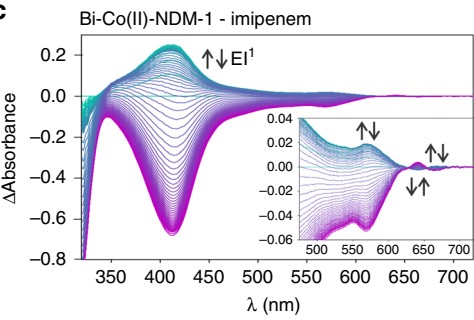

**Fig. 4** Electronic absorption spectra of carbapenem hydrolysis catalysed by binuclear MβLs. **a** Sequence of difference spectra upon the reaction of 150 μM imipenem and 95 μM bi-Zn(II)-NDM-1. The reaction progresses from *orange* to *blue* spectra. The time interval covers from 0.002 to 0.07 s. **b** Sequence of difference spectra upon the reaction of 100 μM meropenem and 100 μM bi-Zn(II)-NDM-1. The reaction progresses from *blue* to *black* spectra. The time interval covers from 0.002 to 0.05 s. **c** Sequence of difference spectra upon the reaction of 450 μM imipenem and 112.5 μM bi-Co(II)-NDM-1. The reaction progresses from *green* to *purple* spectra. The time interval covers from 0.002 to 0.67 s. The *inset* shows a magnification of the 450–720 nm region

geometry are evidenced upon formation of the Michaelis complex, and of intermediate $EI^2$ (Fig. 7a). Substrate binding takes place by expansion of the coordination sphere of $Zn_2$ in mono-Zn (II)-GOB-18, as revealed by XAS data and by the time evolution of the ligand field bands in the Co(II)-substituted enzyme. These observations are consistent with substrate binding to the metal ion through the carboxylate group at C-3, without dissociation of the metal-bound water molecule (ES in Fig. 7a). In contrast, the mono-Zn(II) B2 enzyme Sfh-I did not exhibit significant changes in its coordination geometry during turnover. In this case, it is likely that substrate binding takes place with dissociation of the metal-bound water, maintaining a tetrahedral coordination sphere.

In contrast to the case of mono-Zn(II) enzymes (Sfh-I and GOB-18), where the Michaelis complexes are detectable during the reaction, the concentration of the ES complex has already

decayed within the dead time of the experiments with bi-Zn(II) enzymes. We propose that substrate binding in bi-Zn(II) enzymes takes place also by coordination of the carboxylate group at C-3 to $Zn_2$, with detachment of the bridging hydroxide from this metal site, thus giving rise to a potent nucleophile (ES in Fig. 7b). A terminal hydroxide is expected to be a more efficient nucleophile than a bridging hydroxide, as it is also the case for other metallohydrolases. We thus favour the same hypothesis in this mechanism[40].

The lack of accumulation of the Michaelis complex in the reaction catalysed by bi-Zn(II) enzymes reveals that the step involving the nucleophilic attack (ES → EI[1]) is faster in these enzymes compared to mono-Zn(II) species. The nucleophile in bi-Zn(II) MβLs is the hydroxide moiety bound to the $Zn_1$ ion that is lacking in the mono-Zn(II) variants[10, 22]. Therefore, we conclude that mono-Zn(II) enzymes do not use a metal-activated nucleophile (Fig. 7a), supporting the proposal that a hydrogen bond network activates the nucleophilic water in mono-Zn(II) enzymes[17, 41]. QM-MM calculations and the kinetic data suggest that the two productive intermediates EI[1] and EI[2] correspond to EI$^{WAT}$ and EI, respectively (Fig. 6), both delocalized anionic species. The presence of a water molecule in EI[1] leads us to conclude that, in binuclear enzymes, substrate binding takes place without replacement of the solvent molecule bound to $Zn_2$ in the resting state, as observed in mononuclear GOB-18 (Fig. 7). This intermediate accumulates more in bi-Zn(II) enzymes (Fig. 4), while EI[2] shows higher levels in mono-Zn(II) variants (Fig. 2). These results are consistent with the observation of reaction intermediates at 390 and 380 nm in SPM-1[42] and IMP-25[43], that we are now able to assign to EI[1]. Other enzymes, such as mono-Zn(II) Bla2 did not show accumulation of any intermediate[31].

Formation of intermediate species in bi-Zn(II)-BcII is accompanied by a significant lengthening of the Zn(II)–Zn(II) distance, which finally relaxes to an intermediate distance in the EP adduct. This behaviour resembles that reported for nitrocefin hydrolysis by the binuclear B3 enzyme L1[44], revealing similarities in the ways that MβLs with different active sites stabilise similar reaction intermediates on reaction with different substrates. The study of bi-Co(II)-NDM-1 indicates that, compared to the resting state enzyme, coordination geometry increases in EI[1] and EI[2]. Rapid-freeze-quench EPR studies on bi-Co(II)-BcII and bi-Co(II) L1 have also revealed changes in the metal geometry in the intermediates[28, 45]. MCD spectroscopy could also be exploited with this goal[46]. In EI[1], the deprotonated hydrolysed-carbapenem binds to $Zn_1$ through the C-7 carboxylate and to $Zn_2$ through the C-3 carboxylate and the N atom. In EI[2], the carboxylate group at C-7 becomes a bridging ligand since it is also coordinated to $Zn_2$ (Figs. 6, 7). The proposed coordination spheres of EI[1] and EI[2] in the mono-Zn(II) enzymes are equivalent to those proposed for $Zn_2$ in bi-Zn(II) enzymes (Fig. 7a), further supporting the mechanistic resemblance to the atomic level.

Protonation at the nitrogen atom in EI[1] elicits product formation and dissociation, restoring the active site configuration which, in the case of binuclear enzymes, involves the nucleophilic hydroxide (Fig. 7). Instead, protonation at C-2 in EI[2] leads to accumulation of an EP complex lacking a metal-bound water molecule, in agreement with crystallographic evidence[32, 33]. Restoring the active site configuration takes place after product dissociation. All data here presented strongly support that EI[2] undergoes a stereoselective protonation at C-2 involving a water molecule not bound to the metal site. This mechanistic scheme is fully consistent with all previous experimental evidence[22, 23, 28, 29, 42, 43, 45]. Finally, this mechanism highlights the difference between carbapenem and cephalosphorin hydrolysis. Hydrolysis of cephalosporins with poor leaving groups leads to C-protonation by the α face[30]. This observation is in line with a

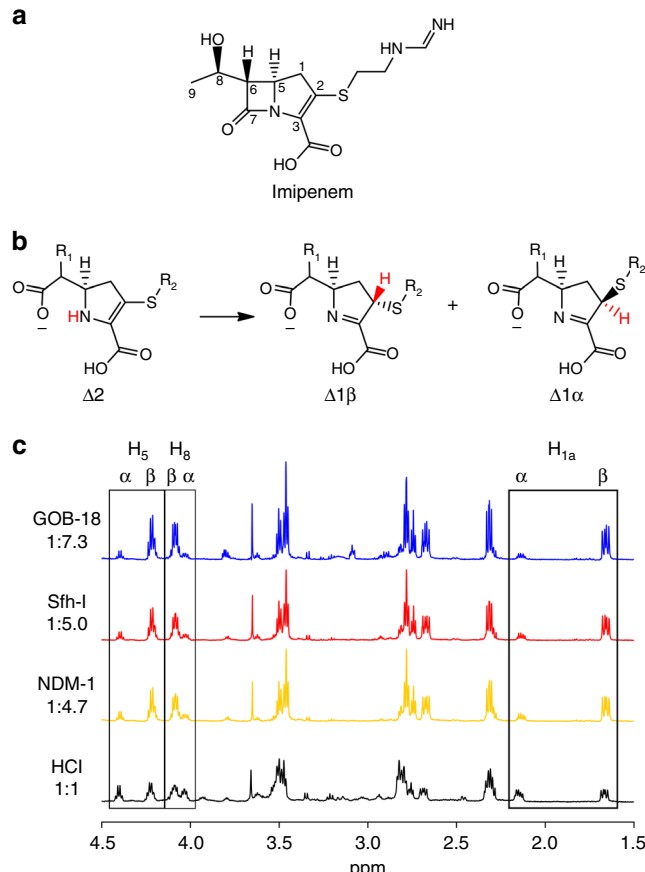

**Fig. 5** Analysis of the population of tautomers in the MβL-catalysed imipenem hydrolysis. **a** Structure and atom numbering of imipenem. **b** Tautomeric structures of hydrolysed imipenem. **c** ¹H-NMR spectra of the products of imipenem hydrolysis by mono-Zn(II)-GOB-18 (*blue*), mono-Zn (II)-Sfh-I (*red*), bi-Zn(II)-NDM-1 (*yellow*), and HCl (*black*). In all cases only tautomer Δ1 was detected. Signal assignments are indicated on top and were taken from Ratcliffe et al.[39]. The α:β diastereomer ratio is indicated on the *left*

recent theoretical study suggesting a different proton donor in cephalosporin hydrolysis, that remains to be tested[47]. The identity of the distinct proton donors may be ultimately assessed by time-resolved crystallography studies.

The role and essentiality of the two Zn(II) ions in MβLs has been a matter of intense controversy[22, 23, 31, 41, 48]. Our study shows that MβLs from different subclasses and with distinct metal contents display a similar catalytic mechanism. This mechanism identifies $Zn_2$ (present in mononuclear and binuclear MβLs, Fig. 1) as playing a central role in substrate binding[26], in providing electrostatic stabilisation for the negative charge of two ring opened anionic intermediate species[28, 29], and in activating the proton donor in one of the productive branches. These mechanistic results are in agreement with recent findings showing that B1 enzymes require a binuclear Zn(II) site in the periplasm to provide resistance[23], and that improvements in the affinity for the $Zn_2$ ion improve fitness[49, 50]. We propose two strategies for the first step involving the nucleophilic attack: a terminal hydroxide bound to the $Zn_1$ ion in binuclear enzymes, or a water molecule activated by a hydrogen bond network in the mononuclear enzymes lacking $Zn_1$, such as Sfh-I and GOB.

The rational design of a "pan-MβL inhibitor" effective against all such enzymes can be envisaged by exploiting these mechanistic features that are common across the full range of enzymes and

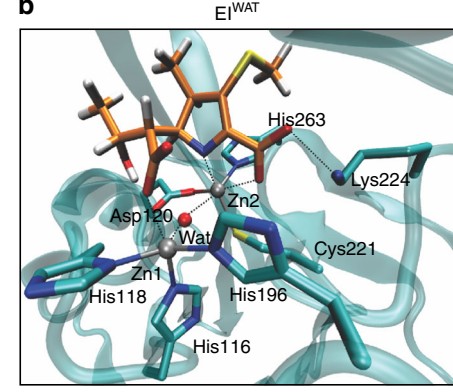

**Fig. 6** Structure of the proposed intermediate species formed during the hydrolysis of carbapenems by bi-Zn(II)-MβLs. Optimized structures of EI **a** and EI^WAT **b**. The quantum protein subsystem is depicted with the conventional liquorice colours (C *cyan*, H *white*, O *red*, N *blue*, and S *yellow*). Zn(II) ions are shown as *grey spheres*. Carbon atoms of the antibiotic derived ligand are shown in *orange*. Coordination bonds of the amino acid residues to Zn(II) are shown with solid lines and interactions with the antibiotic derived ligand are shown with *dotted lines*

are independent of the active site structures and metal content. We recently showed that a minimalistic scaffold mimicking a β-lactam substrate can be employed as an efficient inhibitor of MβLs from all subclasses[51, 52]. Efforts to design MβL inhibitors inspired in the chemical features of these common reaction intermediates are currently underway.

## Methods
**Protein production.** Mono-Zn(II)-GOB-18 and bi-Zn(II)-BcII were expressed as N-terminal fusions to glutathione-S-transferase (GST) protein and purified by using an affinity column with a glutathione-agarose resin. GST was removed by treatment with bovine plasma thrombin (Sigma), and the lactamases were finally purified by ion exchange through a Sephadex CM-50 column[13, 19]. Mono-Zn(II)-Sfh-I was expressed in a pET-26b plasmid (Novagen) and purified by anion exchange (Q-Sepharose) and size exclusion (Superdex 75) chromatography[17]. Bi-Zn(II)-NDM-1 was expressed in a modified version of the pET-28 ( + ) plasmid in which the thrombin cleavage site was replaced by a TEV cleavage site[51]. The crude extract was loaded onto a Ni-Sepharose column, the obtained fusion protein was treated with TEV protease and the mixture was then loaded again onto a Ni-Sepharose column to obtain the pure cleaved protein[51]. All protein preparations have a purity > 95%, as determined by SDS-PAGE. Co(II) substituted enzymes were produced by adding CoSO4 to the apoenzymes, that were obtained by extensive dialysis against buffer (10 mM Hepes, pH 7.5, 200 mM NaCl) containing 20 mM EDTA and Chelex 100[53].

**Rapid kinetics experiments**. The hydrolysis of imipenem catalysed by mono-Zn (II)-GOB-18, mono-Co(II)-GOB-18, mono-Zn(II)-Sfh-I, bi-Zn(II)-NDM-1, and bi-Co(II)-NDM-1 was followed employing a stopped-flow equipment (Applied Photophysics SX.18-MVR) coupled to a photodiode array. The pathlength was 1.0 cm and the integration time was 1.28 s. GOB-18 measurements were performed in 100 mM Hepes, pH 7.5, 200 mM NaCl, at 4 °C, and under pre-steady-state conditions. Measurements with mono-Zn(II)-Sfh-I were performed in 50 mM Hepes, pH 7, at 4 °C, and under pre-steady-state conditions. For NDM-1 the reaction buffer was 100 mM Hepes, pH 7.5, 200 mM NaCl and 300 μM ZnSO4 or 2 equivalents of CoSO4 in the case of bi-Co(II)-NDM-1, the reactions were performed at 6 °C and under pre-steady-state conditions.

Traces obtained at different wavelengths were subject to simultaneous global fit to different kinetic models using the software DynaFit[54]. The molar extinction coefficients employed for mono-Zn(II)-GOB-18 and mono-Co(II)-GOB-18 were: $\varepsilon_{E\ 300\ nm} = 3900\ M^{-1}\ cm^{-1}$ and $\varepsilon_{E\ 340\ nm} = 670\ M^{-1}\ cm^{-1}$ for the free enzyme, $\varepsilon_{S\ 300\ nm} = 9360\ M^{-1}\ cm^{-1}$ and $\varepsilon_{S\ 340\ nm} = 410\ M^{-1}\ cm^{-1}$ for the substrate, $\varepsilon_{ES\ 300\ nm} = 13,260\ M^{-1}\ cm^{-1}$ and $\varepsilon_{ES\ 340\ nm} = 1080\ M^{-1}\ cm^{-1}$ for the complex enzyme-substrate; $\varepsilon_{EI1\ 300\ nm} = 3900\ M^{-1}\ cm^{-1}$ and $\varepsilon_{EI1\ 340\ nm} = 670\ M^{-1}\ cm^{-1}$ for the complex enzyme-intermediate 1 (the same as the free enzyme); $\varepsilon_{EP\ 300\ nm} = 4400\ M^{-1}\ cm^{-1}$ and $\varepsilon_{EP\ 340\ nm} = 825\ M^{-1}\ cm^{-1}$ for the complex enzyme-product; and $\varepsilon_{P\ 300\ nm} = 500\ M^{-1}\ cm^{-1}$ and $\varepsilon_{P\ 340\ nm} = 155\ M^{-1}\ cm^{-1}$ for the product. The molar extinction coefficients employed for mono-Zn(II)-Sfh-I were: $\varepsilon_{E\ 300\ nm} = 5600\ M^{-1}\ cm^{-1}$ and $\varepsilon_{E\ 340\ nm} = 350\ M^{-1}\ cm^{-1}$; $\varepsilon_{S\ 300\ nm} = 9360\ M^{-1}\ cm^{-1}$ and $\varepsilon_{S\ 340\ nm} = 410\ M^{-1}\ cm^{-1}$; $\varepsilon_{ES\ 300\ nm} = 14,960\ M^{-1}\ cm^{-1}$ and $\varepsilon_{ES\ 340\ nm} = 760\ M^{-1}\ cm^{-1}$; $\varepsilon_{EI1\ 300\ nm} = 5600\ M^{-1}\ cm^{-1}$ and $\varepsilon_{EI1\ 340\ nm} = 350\ M^{-1}\ cm^{-1}$ (the same as the free enzyme); $\varepsilon_{EP\ 300\ nm} = 6100\ M^{-1}\ cm^{-1}$ and $\varepsilon_{EP\ 340\ nm} = 505\ M^{-1}\ cm^{-1}$; and $\varepsilon_{P\ 300\ nm} = 500\ M^{-1}\ cm^{-1}$ and $\varepsilon_{P\ 340\ nm} = 155\ M^{-1}\ cm^{-1}$. The molar extinction coefficients employed for bi-Zn(II)-NDM-1 were: $\varepsilon_{E\ 300\ nm} = 4937\ M^{-1}\ cm^{-1}$ and $\varepsilon_{E\ 390\ nm} = 46\ M^{-1}\ cm^{-1}$; $\varepsilon_{S\ 300\ nm} = 7210\ M^{-1}\ cm^{-1}$ and $\varepsilon_{S}$

$_{390\ nm} = 67\ M^{-1}\ cm^{-1}$; $\varepsilon_{EP\ 300\ nm} = 5568\ M^{-1}\ cm^{-1}$ and $\varepsilon_{EP\ 390\ nm} = 86\ M^{-1}\ cm^{-1}$; and $\varepsilon_{P\ 300\ nm} = 630\ M^{-1}\ cm^{-1}$ and $\varepsilon_{P\ 390\ nm} = 40\ M^{-1}\ cm^{-1}$. The molar extinction coefficients employed for bi-Co(II)-NDM-1 were: $\varepsilon_{E\ 412\ nm} = 409\ M^{-1}\ cm^{-1}$, $\varepsilon_{E\ 567\ nm} = 295\ M^{-1}\ cm^{-1}$ and $\varepsilon_{E\ 642\ nm} = 239\ M^{-1}\ cm^{-1}$; $\varepsilon_{S\ 412\ nm} = 15\ M^{-1}\ cm^{-1}$, $\varepsilon_{S\ 567\ nm} = 0\ M^{-1}\ cm^{-1}$ and $\varepsilon_{S\ 642\ nm} = 0\ M^{-1}\ cm^{-1}$; and $\varepsilon_{P\ 300\ nm} = 630\ M^{-1}\ cm^{-1}$, $\varepsilon_{P\ 567\ nm} = 0\ M^{-1}\ cm^{-1}$ and $\varepsilon_{P\ 642\ nm} = 0\ M^{-1}\ cm^{-1}$. $\varepsilon_E$, $\varepsilon_S$, and $\varepsilon_P$ at different wavelengths were determined in each case by measuring the absorbance of different enzyme, substrate or product dilutions at the corresponding wavelength. Then, the molar extinction coefficients were determined by the linear fit of the Lambert-Beer law to absorbance vs. concentration plots. Product samples were obtained by hydrolysing known amounts of the substrate with catalytic concentrations of enzyme. The molar extinction coefficients of enzyme complexes were calculated as the addition of those corresponding to the species interacting in each case.

GOB-18 samples were quantified by using $\varepsilon_{E\ 280\ nm} = 32,200\ M^{-1}\ cm^{-1}$, while $\varepsilon_{E\ 280\ nm} = 35,995\ M^{-1}\ cm^{-1}$ was employed for mono-Zn(II)-Sfh-I, $\varepsilon_{E\ 280\ nm} = 27,960\ M^{-1}\ cm^{-1}$ for NDM-1 and $\varepsilon_{E\ 280\ nm} = 30,500\ M^{-1}\ cm^{-1}$ for bi-Zn(II)-BcII. Substrate samples were quantified by analysing the change in the absorbance at 300 nm due to complete hydrolysis by catalytic amounts of enzyme, and using $\Delta\varepsilon_{300\ nm} = -9000\ M^{-1}\ cm^{-1}$ for imipenem and $\Delta\varepsilon_{300\ nm} = -6500\ M^{-1}\ cm^{-1}$ for meropenem.

**XAS experiments**. For EXAFS studies, resting Sfh-I samples (~1 mM) were supplemented with 20% v/v glycerol as a glassing agent. Product samples were prepared by incubating 0.5 mM Sfh-I with 0.5 mM imipenem, supplemented with 20% (v/v) glycerol, for one hour on ice. EXAFS samples were loaded in Lucite cuvettes with 6 μm polypropylene windows, flash-frozen and stored in liquid nitrogen. Freeze-quenched EXAFS samples were obtained using a modified Update Instruments (Madison, WI) rapid-freeze-quench (RFQ) system[55]. All enzyme and substrate starting concentrations were 1 mM, in 50 mM HEPES metal-free (with Chelex 100, Bio-Rad), pH 7.0, 20 % v/v glycerol. The Update Instrument syringes were driven by a ram connected to a PMI-Kollmorgen stepping motor (model 00D12F-02001-1), which was in turn driven by a model 715 Update Instruments ram controller. The syringes, mixer, and tubing were maintained at 2 °C, in a watertight bath. Immediately prior to sample collection, the nozzle, and the attached mixer for the shortest reaction times, were removed from the bath and placed 5 mm above the surface of 2-methylbutane, contained in a collecting funnel. 2-methylbutane was kept at –130 °C by a surrounding bath (Update Instruments) of liquid nitrogen. Samples were packed into home-designed EXAFS sample holders at –130 °C; excess 2-methylbutane was decanted. All samples were kept in liquid nitrogen until data collection. Calibration of the RFQ system was accomplished by comparing the development of a low-spin Fe(III) EPR signal and the disappearance of a high-spin Fe(III) EPR signal with the associated optical changes at 636 nm, monitored by stopped-flow spectrophotometry, upon mixing excess sodium azide with myoglobin. The shortest, total effective reaction time achieved with the RFQ system was 10 ms[45].

EXAFS spectroscopy: a Si (111) double-crystal monochromator was used at the National Synchrotron Light Source (NSLS), beamline X3B, to measure X-ray absorption spectra, and a Ni mirror was used to accomplish harmonic rejection. Fluorescence excitation spectra for all samples were recorded with a 31-element solid-state Ge detector array. Samples were held at ca 15 K in a Displex cryostat. EXAFS data collection and reduction were performed according to published procedures[55]. Data were measured in duplicate, on two independently prepared samples, by measuring six scans for each sample. Equivalent fits were obtained for the two data sets. The experimental spectra presented are the averaged data sets

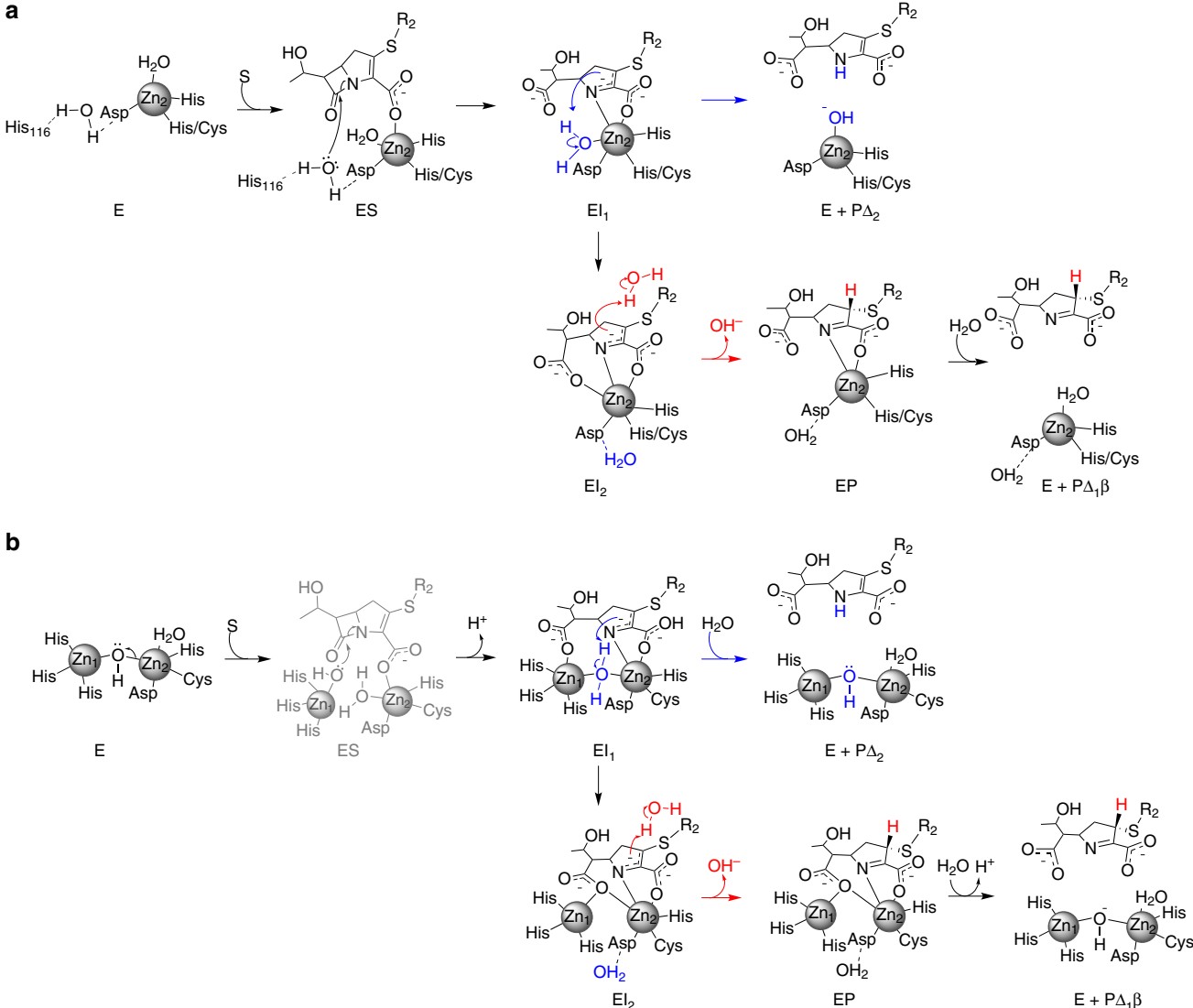

**Fig. 7** Proposed mechanisms for carbapenem hydrolysis by MβLs. **a** Proposed mechanism for mono-Zn(II)-MβLs. **b** Proposed mechanism for bi-Zn(II)-MβLs. The ES complex does not accumulate and is hence depicted in a lighter colour (*grey*). Its structure is proposed based on the ES detected for the mononuclear MβLs

(12 scans per sample). The data were converted from energy to k-space using E0 = 9680 eV.

The nonlinear least-squares engine of IFEFFIT was used to fit Fourier-filtered EXAFS data. IFEFFIT is open source software available from http://www.cars9.uchicago.edu/ifeffit, which is distributed with SixPack; which is available free of charge from http://www.sssrl.slac.stanfrd.edu/_swebb/index.html. Fourier-filtered EXAFS data were fitted utilising theoretical amplitude and phase functions calculated with FEFF v.8.00[56]. The zinc–nitrogen and zinc–sulphur scale factors and the threshold energy, ΔE0, were held fixed in all fits to the data for BcII, GOB, and Sfh-I samples (Zn–N Sc = 0.78, Zn–S Sc = 0.91, ΔE0 = −21 eV, as described previously). First shell fits were first obtained for all reasonable coordination numbers, including mixed nitrogen/oxygen/sulphur ligation, allowing only for variation of the absorber-scattered distances, Ras, and Debye–Waller factors, σas2. Multiple scattering contributions from histidine ligands and metal–metal scattering were fitted according to published procedures[55].

**¹H-NMR measurements and analysis**. The products resulting from the hydrolysis of imipenem catalysed by mono-Zn(II)-GOB-18, mono-Zn(II)-Sfh-I, bi-Zn(II)-NDM-1 and HCl were monitored by ¹H-NMR. A solution of 2 mg/ml imipenem in 100 mM sodium phosphate pH 7.0 at 100% D₂O was incubated at 25 °C with 10 nM of enzyme or HCl until the complete hydrolysis of imipenem. A volume of 500 μl of the mix were placed in an NMR tube and the 1D NMR spectra were collected. All spectra were acquired on Bruker Avance 600 MHz spectrometer equipped with a TXI probe, at 25 °C, with accumulation of 64 scans. We used a sweep width of 10,204 Hz for 1D ¹H-NMR experiments. The excitation sculpting scheme was used

to achieve water suppression[57]; the water-selective 180 °C sine-shaped pulse was 2 ms long. The FID was collected in 32 K data points. Prior to Fourier transformation, a 1 Hz exponential line broadening function was applied. TopSpin 3.0 was used to process and analyse NMR spectra. Signals were assigned based on the work of Ratcliffe et al.[39]. The same scaling ratio was used to plot all spectra in a given series.

**QM-MM calculations**. For hybrid QM–MM calculations[58], the QM subsystem was treated at the density functional level using the programme SIESTA[59]. Basis sets of double-f plus polarisation quality were employed for all atoms, with a pseudoatomic orbital energy shift of 30 meV and a grid cutoff of 150 Ry. Basis sets of double zeta plus polarisation quality were employed for all atoms in the QM subsystem, and all calculations were performed using the generalized gradient approximation functional proposed by Perdew, Burke, and Ernzerhof (PBE)[60]. The Amber99 force field parametrization was used to treat the classical subsystem[61].

The initial structures were taken from the experimental X-ray data: pdb code 4rbs and 4eyl[32]. Hydrogen atoms were added using Amber14 leap module[62]. Both Zn ions (with or without OH⁻/water bridge) plus the coordinated side chains of residues His116, His118, Asp120, His196, Cys221, His263, and the hydrolysed meropenem modified in silico define the quantum subsystem, which comprises in total 82 atoms, also we added Lys224 in some calculations to study the effect of this residue. The rest of the protein and the water molecules ( ~ 23,500 atoms) were treated classically. The simulations were performed by assuming a pH value of 7.5. With the aim of generating the different species (EI¹, EI², EP¹, and EP²), we modified in silico the quantum subsystem by adding or removing hydrogens atoms

or water molecules. EI[2] was obtained as we described above, then we added manually the H atom to C-2 to generate EP[2]. In the case of EI[1] we used the initial structure of EI[2] and we located a water molecule between the zinc ions, then we removed one of the protons of the water molecule and added it to the N atom of the hydrolysed substrate to obtain the initial structure of EP[1]. This method has been successfully applied for the study of metallo-proteins[63–65].

**Molecular dynamics simulations**. The simulation was performed with Amber14 package[62], starting from the crystal structure of NDM-1 bound to hydrolysed meropenem. The system was immersed in a truncated octahedral periodic box with a minimum solute-wall distance of 8 Å, filled with explicit TIP3P water molecules[66]. Ewald sums for treating long-range electrostatic interactions[67]. The SHAKE algorithm was applied to all hydrogen-containing bonds[68]. We used the ff99SB force field implemented in Amber14 to describe the protein. The force field of the active site (Zn, ⁻OH, Asp, Cys, and His) was taken from the literature[69]. The charges and parameters of the hydrolysed substrate were determined using ab initio methods. The van der Waals radius, force constants and equilibrium distances, angles and dihedral were taken from gaff database[62]. Partial charges were RESP charges computed using Hartree–Fock method and 6-31 G* basis set[70]. The temperature and pressure were controlled by the Berendsen thermostat and barostat, respectively, as implemented in Amber14[62]. Cutoff values used for the van der Waals interactions were 10 Å. The system was equilibrated at 300 K using a conventional protocol[49] and then subjected to 10 ns of simulation in the NVT ensemble. We applied a restraint to keep the hydrolysed meropenem bound to the active site.

**Data availability**. Supporting data are available from the corresponding authors upon reasonable request.

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

## Acknowledgements

M.-N.L., A.R.P., and M.M.G. were recipients of doctoral fellowships from CONICET. A.J.V., D.M.M., and L.I.L. are staff members from CONICET. This work was supported by grants from ANPCyT and NIH (R01 AI100560-01) to A.J.V., and from NIH (GM093987) and the National Science Foundation (CHE-1151658) to M.W.C. and D.L.T.; We also acknowledge P30-EB-009998 to the Center for Synchrotron Biosciences from the NIBIB, which supports beamline X3B at the NSLS.

## Author contributions

M.-N.L. and A.R.P. purified protein samples, performed and analysed the stopped-flow experiments. M.M.G. purified protein samples and ran the NMR spectra. D.M.M. performed the QM-MM and MD calculations. M.A., M.W.C., and D.L.T. ran the XAS experiments. M.-N.L., A.R.P., D.L.T., R.A.B., J.S., L.I.L., and A.J.V. analysed and discussed data. M.-N.L., A.R.P., L.I.L., and A.J.V. wrote the paper, and all authors discussed the results and commented on the manuscript.

## Additional information

**Competing interests:** The authors declare no competing financial interests.

