## [Peer Review File · Nature Communications]

Reviewers' comments:

Reviewer #1 (Remarks to the Author):

Metallo-beta-lactamases are clinically important enzymes due to their wide spread in bacterial pathogens and ability to confer resistance to beta-lactam antibiotics, including drugs of last resort, carbapenems. Based on structural architecture and substrate profiles metallo-beta-lactamases are divided into three subclasses, B1, B2 and B3, which can further be subdivided into mononuclear and binuclear enzymes based on the number of zinc atoms in their active site. Active sites of metallo-beta-lactamases are structurally diverse which complicates elucidation of their catalytic mechanism and design of inhibitors for these enzymes.

The authors of the submitted manuscript utilized stopped-flow kinetics, various spectroscopic techniques and x-ray crystallography to study a series of metallo-beta-lactamases. Based on the results of these experiments and data accumulated prior to this study, they proposed that all metallo-beta-lactamases, both mononuclear and binuclear, utilize a common catalytic mechanism. Knowledge of a common mechanism opens new avenues for the rational design of inhibitors for these clinically important enzymes. The manuscript is very well written, the experimental approach is solid and well presented.

I have only one minor comment; on page 6 lines 119-121 it is stated that "The subclass B3 GOB enzymes have Gln residue at position 116...", however on Figure 1 Gln is incorrectly abbreviated as N instead of Q. Please correct.

Reviewer #2 (Remarks to the Author):

I really enjoyed reading this manuscript which provides new mechanistic insights into the mechanism of metallo- β -lactamases (MBLs), enzymes with increasing importance in antibacterial resistance. Although several papers, including some from the same group have already reported studies on the mechanism of MBLs, the new work does provide new insights. Therefore, I would recommend to be published, after minor revision, as the manuscript can still be improved. The authors most likely have also considered doing time resolved studies e.g. time resolved crystallography and I am convinced that they would give more experimental details that can back-up the QM-MM calculations, but these are not essential at this stage. There are some minor errors that can be fixed, and some minor experiments that should be done before publishing this manuscript.

Specific points are addressed below:

The title "A General Reaction Mechanism for Mono- and Binuclear Metallo- β -lactamases" should be revised to include that the work is based on understanding the mechanism of hydrolysis of carbapenems, not all β -lactam antibiotics e.g. cephalosporins, this can mislead the readers, so more specificity is recommended

Abstract,

- line 31 "Inhibitors of MBLs are currently unavailable as design has been limited by the structural diversity of their active sites"; cyclic boronates are currently in phase I clinical trials see NCT02955459, moreover other inhibitors of MBLs are under development, including maybe by the authors, therefore this should be rewritten maybe as "no clinically approved, etc."

Introduction

- line 51-53 "Carbapenem-resistant Gram-negative bacteria are rapidly emerging as a cause of opportunistic healthcare-associated infections and can give rise to mortality of almost 50% of the infected patients." I'm not sure on this, it sounds slightly as a bold statement could you please add relevant references to this statement?

- line "Metallo- β -lactamases (M β Ls) are the largest and most efficient family of carbapenemases" again I feel this is a slightly big statement; in the SI the authors make an analysis and show that e.g. the Class A and D carbapenemases together are a lower number than Class B MBLs so the authors could say "one of the largest etc."
- line 85 "the identity of the nucleophile, which is the Zn1-bound hydroxide in bi-Zn(II) enzymes" similarly needs to be rephrased e.g. "... which was proposed to be the Zn-1 bound..." The identity of the nucleophile has not been elucidated yet, thus either the authors conduct e.g. time resolved studies or this needs to be tuned down.
- similarly in line 103 "The proposed structures for these intermediates allowed us to identify the proton donors in this mechanism." What experiments have the authors performed to be confident about the identity of the proton donor? Time resolved studies need to be conducted to have such a statement or the authors need to say "suggested".

Results

- line 166 - "mono-Co(II)-GOB-18" the authors should show in the SI the native MS spectra for the mono-Co(II)-GOB-18 (and before that the mono-Zn-GOB), this can be really useful for the reader
- the authors should add an extra table (e.g. SI) that summarises the wavelength for each species observed, this can be helpful for the reader

Section "Identity of the reaction intermediates and of the product complexes" the authors should update this section; in the current form of the manuscript they only used the crystal structures of NDM-1 with carbapenems and did not mention the SMB-1 (B3 MBL) carbapenems crystal structures (see *Antimicrob Agents Chemother.* 2016 Jun 20;60(7):4274-82. doi: 10.1128/AAC.03108-15); if all B1, B2 and B3 have the same mechanism as the authors claim this needs to be included/reanalysed. As a note the proposed mechanism of SMB-1 based on the crystal structure resembles the mechanism that the authors describe in the current manuscript. Also in the same section the authors do not comment on that MBL catalysed cephalosporin hydrolysis yields to α -diastereomers - ref 31. I would also recommend to test cephalosporin hydrolysis similarly as described in ref 31, using NDM-1 and the same experimental set up used for carbapenems and compare it, this should not take too much effort.

Figure 3 - EI1 and EI2, the numbers should be superscript.

SI -Fig 15 - Imipenem shouldn't that say meropenem?

Overall, I do believe the work is of sufficiently broad interest to merit publication after the minor revisions suggested.

Reviewer #3 (Remarks to the Author):

Dear authors and editor,

I greatly enjoyed reading this manuscript describing a detailed functional characterization of the mechanism of metallo-beta-lactamases (MBLs). I have a few comments/suggestions (listed below) that reflect my great interest in this paper, and which were added with the intention to help the authors to widen the scope and impact of their study. However, I like to point out that I find the paper in its present form adequate for publication in *Nature Communications* (assuming that at least Points 1 & 6 below will be addressed adequately). I like to congratulate the team for an impressive approach involving a range of complementary techniques to substantiate a model for a catalytic mechanism that may be common to all members of the large family of MBLs.

Points to consider:

1) On page 6/7 (Results & Discussion) the branched model for the mechanism is introduced, involving two intermediate states, EI1 and EI2 (the numbers should probably be in superscript in Figure 3). The presence of two intermediate states is interesting and well-founded based on their presented experimental data. This interpretation might also be related to previous observations reported from fast-kinetics studies with other MBLs (albeit with different substrates). Specifically, I'm referring to Bla2, where stopped-flow measurements do not display the presence of an intermediate, and others (e.g. L1, BcII), where similar measurements do demonstrate the presence of at least one intermediate (compare Hawk et al (2009) JACS 131, 10753 and Garrity et al (2005) Biochemistry 44, 1078). In a recent study it was then shown that the B3-type MBL AIM-1 may be able to promote two alternative catalytic pathways, one related to that of Bla2, and the other similar to that of L1, etc (see Selleck et al (2016) Chem Eur J 22, 17704). I thus wonder if in particular the reaction with AIM-1 also demonstrates the presence of a branched mechanism. Could these earlier results also be related to two alternative protonation routes, as described by the authors on page 14? Apart from being strongly interested in hearing the authors' expert opinion about this suggestion I also feel that incorporating a small discussion of how their interpretation relates to these previous studies may enhance the significance of their work.

2) The authors generated the Co(II)-derivatives of some of the enzymes in their paper. While some comments about the impact of the metal replacement on catalytic properties were made it may be informative to make relevant information more readily accessible (i.e. give numbers of k_{cat} and k_{cat}/K_m values for the Zn(II) and Co(II)-derivatives, where available). Does replacement of the native Zn(II) by Co(II) have a "conserved" effect among various MBLs?

3) Especially since the authors investigated Co(II)-derivatives of MBLs it would be very interesting to employ magnetic circular dichroism (MCD) to probe the ligand environment. I am not suggesting that the authors ought to include such data in the present study but in their final conclusions they may allude to possible future studies to probe the effect of substrate or inhibitor binding using MCD as described in recent studies such as the one by Selleck et al mentioned above.

4) Related to the above, is Sfh-I active as a Co(II) derivative?

5) A more technical comment is focused on the optimization of the conditions for the stopped-flow measurements. For instance, in the illustrations shown in Figure 2 the excess of substrate over enzyme varies from ~two-fold for Zn(II)-GOB-18 to nearly ten-fold for Sfh-I. In contrast, for measurements with NDM-1 (Figure 4) the conditions generally resembled single-turnover conditions. It might be informative to comment on how the conditions for individual measurements were optimized (and what criteria were employed).

6) On page 15/16 the authors discuss the mechanism of action. Of particular interest is their proposal that substrate binding may lead to a change of the position of the metal ion-bridging hydroxide to a terminal position. This shift is likely to enhance the nucleophilicity of this hydroxide, thus enhancing reactivity. I am not aware that this possibility has been discussed for MBLs previously (please correct me if I'm wrong) but it certainly has been discussed at length for other metallohydrolases. The authors should thus expand their discussion to correlate their proposal to those discussed previously for purple acid phosphatases or some organophosphate-degrading hydrolases (e.g. Hadler et al (2009) JACS; Mitic et al (2010) JACS and references therein; also see reviews Acc Chem Res (2012), Coord Chem Rev (2013) or Coord Chem Rev (2016) by Schenk and coworkers).

7) In the conclusion I didn't quite understand the reasoning behind the statement starting on line 421 - the authors say that the metal ion in site Zn2 plays an essential role in the mechanism, and that this is in agreement with a study that indicated that B1-type MBLs require a binuclear site in the periplasm to provide resistance. Is the mononuclear form of NDM-1 capable of providing resistance? It appears the metal in the Zn1 site is used mainly to activate the nucleophile.

However, the authors continue by saying that in mononuclear forms hydrogen bonding interactions may be responsible for the activation of the nucleophile, which despite being "the weaker nucleophile activation does not impair the overall performance of the enzymes". How is this performance measured, relative to what standard?

8) A couple of minor points:

a) Materials and Methods - XAS: Since one of the co-authors published a very informative review on EXAFS/XANES data analysis I recommend the inclusion of this article (Tierney, Biophys. J. 107: 1263-1272) as it is a helpful tool for the reader to grasp the full potential of this methodology.

b) Page 8, line 168: Rewrite this sentence - grammatically incorrect.

c) Page 10, line 239: "higher energy" instead of "higher energies".

d) Page 17, line 426: Remove "Instead"

I look forward to reading the comments of the authors and hope to read the final article soon in a future issue of Nat. Comm.

Best wishes,

Gary Schenk

Reviewer #4 (Remarks to the Author):

In the QM/MM part of the manuscript, the analysis in the current form is convincing. But there are few minor concerns:

1. What's the value of pH in the simulation system and the size of the system ? In the method section (Rapid kinetics experiments), the authors used the ions or solutes (Hepes, NaCl ...) why the authors didn't use the solutes in their simulations ?
2. The author did not provide any evidence to show the quality of the simulation.

Response to Referees' Letter

Ref: NCOMMS-17-06972

Title: "A General Reaction Mechanism for Mono- and Binuclear Metallo- β -lactamases"

Authors: Lisa, Palacios et al.

Reviewer #1 (Remarks to the Author):

Metallo-beta-lactamases are clinically important enzymes due to their wide spread in bacterial pathogens and ability to confer resistance to beta-lactam antibiotics, including drugs of last resort, carbapenems. Based on structural architecture and substrate profiles metallo-beta-lactamases are divided into three subclasses, B1, B2 and B3, which can further be subdivided into mononuclear and binuclear enzymes based on the number of zinc atoms in their active site. Active sites of metallo-beta-lactamases are structurally diverse which complicates elucidation of their catalytic mechanism and design of inhibitors for these enzymes.

The authors of the submitted manuscript utilized stopped-flow kinetics, various spectroscopic techniques and x-ray crystallography to study a series of metallo-beta-lactamases. Based on the results of these experiments and data accumulated prior to this study, they proposed that all metallo-beta-lactamases, both mononuclear and binuclear, utilize a common catalytic mechanism. Knowledge of a common mechanism opens new avenues for the rational design of inhibitors for these clinically important enzymes. The manuscript is very well written, the experimental approach is solid and well presented.

I have only one minor comment; on page 6 lines 119-121 it is stated that "The subclass B3 GOB enzymes have Gln residue at position 116...", however on Figure 1 Gln is incorrectly abbreviated as N instead of Q. Please correct.

Answer: We thank the reviewer for making us note this mistake, it has been amended.

Reviewer #2 (Remarks to the Author):

I really enjoyed reading this manuscript which provides new mechanistic insights into the mechanism of metallo- β -lactamases (MBLs), enzymes with increasing importance in antibacterial resistance. Although several papers, including some from the same group have already reported studies on the mechanism of MBLs, the new work does provide new insights. Therefore, I would recommend to be published, after minor revision, as the manuscript can still be improved. The authors most likely have also considered doing time resolved studies e.g. time resolved crystallography and I am convinced that they would give more experimental details that can back-up the QM-MM calculations, but these are not essential at this stage.

There are some minor errors that can be fixed, and some minor experiments that should be done before publishing this manuscript.

Specific points are addressed bellow:

The title "A General Reaction Mechanism for Mono- and Binuclear Metallo- β -lactamases" should be revised to include that the work is based on understanding the mechanism of hydrolysis of carbapenems, not all β -lactam antibiotics e.g. cephalosporins, this can mislead the readers, so more specificity is recommended

Answer: This suggestion is very appropriate. We have modified the title accordingly: "A General Reaction Mechanism for Carbapenem Hydrolysis by Mono- and Binuclear Metallo- β -lactamases"

Abstract,

- line 31 "Inhibitors of M β LS are currently unavailable as design has been limited by the structural diversity of their active sites"; cyclic boronates are currently in phase I clinical trials see NCT02955459, moreover other inhibitors of MBLs are under development, including maybe by the authors, therefore this should be rewritten maybe as "no clinically approved, etc."

Answer: The reviewer is correct. We have not been clear enough.

We have now modified the expression: "Inhibitors of MBLs are currently unavailable..." by "Clinically approved inhibitors of MBLs are currently unavailable..." (page2, line 33)

Introduction

- line 51-53 "Carbapenem-resistant Gram-negative bacteria are rapidly emerging as a cause of opportunistic healthcare-associated infections and can give rise to mortality of almost 50% of the infected patients." I'm not sure on this, it sounds slightly as a bold statement could you please add relevant references to this statement?

Answer: The 50% of mortality corresponds to some CRE, so we have modified the statement accordingly to "with high mortality rates".

- line "Metallo- β -lactamases (M β LS) are the largest and most efficient family of carbapenemases" again I feel this is a slightly big statement; in the SI the authors make an analysis and show that e.g. the Class A and D carbapenemases together are a lower number than Class B MBLs so the authors could say "one of the largest etc."

Answer: We have toned down our statement based on this suggestion (page 3, line 53).

- line 85 “the identity of the nucleophile, which is the Zn1-bound hydroxide in bi-Zn(II) enzymes” similarly needs to be rephrased e.g. “... which was proposed to be the Zn-1 bound...” The identity of the nucleophile has not been elucidated yet, thus either the authors conduct e.g. time resolved studies or this needs to be tuned down.

Answer: We have toned down our statement based on this suggestion. Anyway, we should note that there have been two candidates proposed as possible attacking nucleophiles: the carboxylate group of the conserved Asp120 and the Zn(II)-bound hydroxide ion, and Asp120 was discarded as the nucleophile in early works by Mike Page and Steve Benkovic by sound biochemical experiments (Bounaga et al. (1998), *Biochem. J.* 33, 703-11; Wang et al. (1999) *Biochemistry*, 38, 10013-23). The general consensus accepts that the Zn(II)-bound hydroxide ion is the attacking nucleophile, as it is the case for most Zn(II) hydrolases, in which the metal ion lowers the pKa of the bound water molecule.

- similarly in line 103 “The proposed structures for these intermediates allowed us to identify the proton donors in this mechanism.” What experiments have the authors performed to be confident about the identity of the proton donor? Time resolved studies need to be conducted to have such a statement or the authors need to say “suggested”.

Answer: We have toned down our statement based on this suggestion. Moreover, in the discussion we make reference to the need to perform time-resolved crystallographic studies (page 17; line 429).

Results

- line 166 – “mono-Co(II)-GOB-18” the authors should show in the SI the native MS spectra for the mono-Co(II)-GOB-18 (and before that the mono-Zn-GOB), this can be really useful for the reader

Answer: The metal content in mono-Co(II)-GOB was assessed by atomic absorption spectroscopy or ICP-MS. Determination of the total mass of the metal derivative is of help, but does not address the problem whether the mono-nuclear variant has the metal ion localized in site 1 or site 2. This issue was sorted out by NMR spectroscopy (Moran-Barrio, J. *et al.* (2007) *J Biol Chem* 282, 18286-93), that allowed us to determine that these mono-metallic variants have the metal ion localized in the Zn2 site.

- the authors should add an extra table (e.g. SI) that summarises the wavelength for each species observed, this can be helpful for the reader

Answer: Based on this suggestion, we added a new Table in the Supplementary Information (Supplementary Table 1).

Section “Identity of the reaction intermediates and of the product complexes” the authors should update this section; in the current form of the manuscript they only used the crystal structures of NDM-1 with carbapenems and did not mention the SMB-1 (B3 MBL) carbapenems crystal structures (see *Antimicrob Agents Chemother.* 2016 Jun 20;60(7):4274-82. doi: 10.1128/AAC.03108-15); if all B1, B2 and B3 have the same mechanism as the authors claim this needs to be included/reanalysed. As a note the proposed mechanism of SMB-1 based on the crystal structure resembles the mechanism that the authors describe in the current manuscript.

Answer: This is an excellent suggestion, and we apologize for this involuntary omission. Indeed, the structures of hydrolyzed carbapenems with SMB-1 fully support the mechanism proposed

in our manuscript, both in the proposal of a branched mechanism and in the stereospecificity of C-2 protonation, that takes place on the β face of the carbapenem moiety. We now discuss the results from this article by Wachino and coworkers in the text both in the Introduction (page 4, line 94) and in the Results section (page 14, line 350).

Also in the same section the authors do not comment on that MBL catalysed cephalosporin hydrolysis yields to α -diastereomers – ref 31. I would also recommend to test cephalosporin hydrolysis similarly as described in ref 31, using NDM-1 and the same experimental set up used for carbapenems and compare it, this should not take too much effort.

Answer: We monitored cephalixin hydrolysis by NDM-1 followed by NMR (see below) and we confirmed that indeed, hydrolysis of cephalosporins with poor leaving groups in the C-3 position (such as cephalixin) yields mostly α -diastereomers, as reported not only for NDM-1 but also for *E. cloacae* P99 lactamase (Vilanova *et al.* (1997) *J. Chem. Soc., Perkin Trans. 2*, 2439). These results do not contradict our findings but, instead, show that carbon protonation differs between cephalosporin hydrolysis and carbapenem hydrolysis. Crystallographic evidence supports this conclusion, since the work by Feng and coworkers (ref. 30) clearly shows that cephalixin is C-protonated by the α -face, and the work by Wachino and coworkers in SMB-1 shows that the opposite occurs in carbapenems. Moreover, a recent theoretical study (Das and Nair (2017) *PCCP*, 19, 13111) suggests a different proton donor in cephalosporin hydrolysis. Despite cephalosporin hydrolysis may lead to different products depending on the substituent at C-3, we added a comment in this regard and we thank the reviewer for his/her comment (page 17, line 425).

Blue: Unhydrolysed cephalixin.

Red: Product of the complete hydrolysis of cephalixin by bi-Zn(II)-NDM-1. The alpha-beta ratio was 3.7:1

Figure 3 - E11 and E12, the numbers should be superscript.

Answer: We thank the reviewer for making us note this mistake.

SI -Fig 15 – Imipenem shouldn't that say meropenem?

Answer: We thank the reviewer for making us note this mistake.

Overall, I do believe the work is of sufficiently broad interest to merit publication after the minor revisions suggested.

Reviewer #3 (Remarks to the Author):

Dear authors and editor,

I greatly enjoyed reading this manuscript describing a detailed functional characterization of the mechanism of metallo-beta-lactamases (MBLs). I have a few comments/suggestions (listed below) that reflect my great interest in this paper, and which were added with the intention to help the authors to widen the scope and impact of their study. However, I like to point out that I find the paper in its present form adequate for publication in Nature Communications (assuming that at least Points 1 & 6 below will be addressed adequately). I like to congratulate the team for an impressive approach involving a range of complementary techniques to substantiate a model for a catalytic mechanism that may be common to all members of the large family of MBLs.

Points to consider:

1) On page 6/7 (Results & Discussion) the branched model for the mechanism is introduced, involving two intermediate states, E1¹ and E1² (the numbers should probably be in superscript in Figure 3). The presence of two intermediate states is interesting and well-founded based on their presented experimental data. This interpretation might also be related to previous observations reported from fast-kinetics studies with other MBLs (albeit with different substrates). Specifically, I'm referring to Bla₂, where stopped-flow measurements do not display the presence of an intermediate, and others (e.g. L1, BcII), where similar measurements do demonstrate the presence of at least one intermediate (compare Hawk et al (2009) JACS 131, 10753 and Garrity et al (2005) Biochemistry 44, 1078). In a recent study it was then shown that the B3-type MBL AIM-1 may be able to promote two alternative catalytic pathways, one related to that of Bla₂, and the other similar to that of L1, etc (see Selleck et al (2016) Chem Eur J 22, 17704). I thus wonder if in particular the reaction with AIM-1 also demonstrates the presence of a branched mechanism. Could these earlier results also be related to two alternative protonation routes, as described by the authors on page 14? Apart from being strongly interested in hearing the authors' expert opinion about this suggestion I also feel that incorporating a small discussion of how their interpretation relates to these previous studies may enhance the significance of their work.

Answer: We agree with the reviewer. Since there is a significant amount of mechanistic papers in MBLs, we focused on the Discussion on some of them studying the mechanism of carbapenem hydrolysis. We have now included comments on the paper on Bla₂ (where no intermediate has been found in the hydrolysis of carbapenems by mono-Zn(II) Bla₂ (page 16, line 400), and Co(II)-substituted L1, in which EPR data provides evidence of a similar intermediate reported for Co(II)-substituted BcII (page 17, line 408). Regarding AIM-1, in our opinion, the kinetic data reported in that paper do not suggest the presence of a branched mechanism for the hydrolysis of nitrocefim. A wider range of substrate:enzyme ratios might be explored to assess whether there is a branched mechanism. I would definitively suggest running pre-steady state kinetic experiments of carbapenem hydrolysis by AIM-1 to directly confirm that issue.

2) The authors generated the Co(II)-derivatives of some of the enzymes in their paper. While some comments about the impact of the metal replacement on catalytic properties were made it may be informative to make relevant information more readily accessible (i.e. give numbers of *k*_{cat} and *k*_{cat}/*K*_m values for the Zn(II) and Co(II)-derivatives, where available).

Does replacement of the native Zn(II) by Co(II) have a "conserved" effect among various MBLs?

Answer: This is an important issue, which we unintentionally overlooked. Part of these data was already available in the literature. We have now included a Table in the Supplementary Information reporting these data (Supplementary Table 2).

3) Especially since the authors investigated Co(II)-derivatives of MBLs it would be very interesting to employ magnetic circular dichroism (MCD) to probe the ligand environment. I am not suggesting that the authors ought to include such data in the present study but in their final conclusions they may allude to possible future studies to probe the effect of substrate or inhibitor binding using MCD as described in recent studies such as the one by Selleck et al mentioned above.

Answer: This is a great suggestion. We have added a related comment in the Discussion section (page 17, line 410).

4) Related to the above, is Sfh-I active as a Co(II) derivative?

Answer: Co(II)-Sfh-I is active. The corresponding data have also been included in the Supplementary Table 2.

5) A more technical comment is focused on the optimization of the conditions for the stopped-flow measurements. For instance, in the illustrations shown in Figure 2 the excess of substrate over enzyme varies from ~two-fold for Zn(II)-GOB-18 to nearly ten-fold for Sfh-I. In contrast, for measurements with NDM-1 (Figure 4) the conditions generally resembled single-turnover conditions. It might be informative to comment on how the conditions for individual measurements were optimized (and what criteria were employed).

Answer: We optimized enzyme and substrate concentrations in stopped-flow experiments to detect clear UV-vis absorption signals within the equipment in the linear response range. We needed to use a wide range of concentrations to better trap both reaction intermediates and define the branched mechanism.

6) On page 15/16 the authors discuss the mechanism of action. Of particular interest is their proposal that substrate binding may lead to a change of the position of the metal ion-bridging hydroxide to a terminal position. This shift is likely to enhance the nucleophilicity of this hydroxide, thus enhancing reactivity. I am not aware that this possibility has been discussed for MBLs previously (please correct me if I'm wrong) but it certainly has been discussed at length for other metallohydrolases. The authors should thus expand their discussion to correlate their proposal to those discussed previously for purple acid phosphatases or some organophosphate-degrading hydrolases (e.g. Hadler et al (2009) JACS; Mitic et al (2010) JACS and references therein; also see reviews Acc Chem Res (2012), Coord Chem Rev (2013) or Coord Chem Rev (2016) by Schenk and coworkers).

Answer: We appreciate this comment. In a recent review article (Meini et al. (2015) *FEBS Lett* 589, 3419-32) we have specifically discussed this issue. Anyway, we have modified the discussion including these comments with a comparison to related systems, as suggested by the reviewer (page 16, line 383).

7) In the conclusion I didn't quite understand the reasoning behind the statement starting on line 421 - the authors say that the metal ion in site Zn2 plays an essential role in the

mechanism, and that this is in agreement with a study that indicated that B1-type MBLs require a binuclear site in the periplasm to provide resistance. Is the mononuclear form of NDM-1 capable of providing resistance? It appears the metal in the Zn1 site is used mainly to activate the nucleophile. However, the authors continue by saying that in mononuclear forms hydrogen bonding interactions may be responsible for the activation of the nucleophile, which despite being "the weaker nucleophile activation does not impair the overall performance of the enzymes". How is this performance measured, relative to what standard?

Answer: There are several issues raised by the reviewer that have already been discussed at length in a previous paper from our group (Gonzalez et al. (2012) *Nature Chemical Biology* 8,698-700) and may not require a long consideration in the present paper. The non-metal activated nucleophile correspond to mono-Zn(II) enzymes with the metal ion localized at the Zn2 site (B2 enzymes and mono-Zn(II) GOB in the present work). We have rewritten the final conclusion to make it clearer, and we removed the sentence of the weaker nucleophile, since we considered it confusing.

8) A couple of minor points:

a) Materials and Methods - XAS: Since one of the co-authors published a very informative review on EXAFS/XANES data analysis I recommend the inclusion of this article (Tierney, *Biophys. J.* 107: 1263-1272) as it is a helpful tool for the reader to grasp the full potential of this methodology.

b) Page 8, line 168: Rewrite this sentence - grammatically incorrect.

c) Page 10, line 239: "higher energy" instead of "higher energies".

d) Page 17, line 426: Remove "Instead"

Answer: We appreciate that the reviewer has taken care of all details to help us improve the quality of our manuscript, and we have taken care of these mistakes and suggestions.

I look forward to reading the comments of the authors and hope to read the final article soon in a future issue of *Nat. Comm.*

Best wishes,

Gary Schenk

Reviewer #4 (Remarks to the Author):

In the QM/MM part of the manuscript, the analysis in the current form is convincing. But there are few minor concerns:

1. What's the value of pH in the simulation system and the size of the system ? In the method section (Rapid kinetics experiments), the authors used the ions or solutes (Hepes, Nacl ...) why the authors didn't use the solutes in their simulations ?

Answer: The simulations were performed by assuming a pH value of 7.5. This has been added in the text, page 23, line 576.

The quantum subsystem includes the Zn(II) ions plus the coordinated side chains of residues of the first coordination sphere, and the hydrolyzed meropenem modified in silico which comprises in total 82 atoms, as we now described in methods (page 23, line 574). The rest of the protein and the water molecules were treated classically (~23,500 atoms).

The simulations were typically performed without the explicit consideration of counter ions because of sampling limitations. We performed QM-MM calculations to consider the effect of the protein environment and the water molecules, but we have no experimental evidence suggesting the need to add counter ions in our simulations.

2. The author did not provide any evidence to show the quality of the simulation.

Answer: The simulation performed at the QM-MM level employing DFT for describing the quantum subsystem correspond to state-of-the art calculations and typical setups have been employed successfully in several Works. This methodology is able to capture the main features of the reactive species investigated in this work. (J. Biol. Inorg. Chem. 18, 223-232 (2013); J. Am. Chem.Soc. 128, 12817-12828 (2006); J. Am. Chem.Soc. 127, 7721-7728 (2005); Biochemistry 55, 3403-3417 (2016); Proteins 82, 1004-1021 (2014)).

REVIEWERS' COMMENTS:

Reviewer #2 (Remarks to the Author):

No any further comments. The authors have answered my earlier comments. Nice work!